# Carry-Over of Zearalenone and Its Metabolites to Intestinal Tissues and the Expression of CYP1A1 and GSTπ1 in the Colon of Gilts before Puberty

**DOI:** 10.3390/toxins14050354

**Published:** 2022-05-18

**Authors:** Magdalena Mróz, Magdalena Gajęcka, Paweł Brzuzan, Sylwia Lisieska-Żołnierczyk, Dawid Leski, Łukasz Zielonka, Maciej T. Gajęcki

**Affiliations:** 1Department of Veterinary Prevention and Feed Hygiene, Faculty of Veterinary Medicine, University of Warmia and Mazury in Olsztyn, Oczapowskiego 13/29, 10-718 Olsztyn, Poland; magdalena.mroz@uwm.edu.pl (M.M.); lukaszz@uwm.edu.pl (Ł.Z.); gajecki@uwm.edu.pl (M.T.G.); 2Department of Environmental Biotechnology, Faculty of Environmental Sciences and Fisheries, University of Warmia and Mazury in Olsztyn, Słoneczna 45G, 10-719 Olsztyn, Poland; brzuzan@uwm.edu.pl; 3Independent Public Health Care Centre of the Ministry of the Interior and Administration, and the Warmia and Mazury Oncology Centre in Olsztyn, Wojska Polskiego 37, 10-228 Olsztyn, Poland; lisieska@wp.pl; 4Research and Development Department, Wipasz S.A., Wadąg 9, 10-373 Wadąg, Poland; dawid.leski@wipasz.pl

**Keywords:** zearalenone, low dose, intestines, carry-over, CYP1A1 and GSTπ1, prepubertal gilts

## Abstract

The objective of this study was to evaluate whether low doses of zearalenone (ZEN) affect the carry-over of ZEN and its metabolites to intestinal tissues and the expression of CYP1A1 and GSTπ1 in the large intestine. Prepubertal gilts (with a BW of up to 14.5 kg) were exposed in group ZEN to daily ZEN5 doses of 5 μg/kg BW (*n* = 15); in group ZEN10, 10 μg/kg BW (*n* = 15); in group ZEN15, 15 μg/kg BW (*n* = 15); or were administered a placebo (group C, *n* = 15) throughout the experiment. After euthanasia, tissues were sampled on exposure days 7, 21, and 42 (D1, D2, and D3, respectively). The results confirmed that the administered ZEN doses (LOAEL, NOAEL, and MABEL) were appropriate to reliably assess the carry-over of ZEN. Based on the observations made during 42 days of exposure to pure ZEN, it can be hypothesized that all mycotoxins (ZEN, α-zearalenol, and β-zearalenol) contribute to a balance between intestinal cells and the expression of selected genes encoding enzymes that participate in biotransformation processes in the large intestine; modulate feminization processes in prepubertal gilts; and elicit flexible, adaptive responses of the macroorganism to mycotoxin exposure at the analyzed doses.

## 1. Introduction

Zearalenone is an undesirable substance that is widely encountered in cereal kernels and cereal products [1]. It is produced mainly by *Fusarium graminearum* as a metabolite with estrogenic properties. Pure ZEN is a white crystalline compound with the chemical formula C_18_H_22_O_5_, a molecular mass of 318.36 g/mol, and a melting temperature range of 161–163 °C. Zearalenone is not soluble in water, but it dissolves easily in alkaline solutions, ether, benzene, methanol, and ethanol. This mycotoxin can be metabolized by various organisms during phase I and II biotransformation processes [2]. Feeds contain modified forms of ZEN (ZELs), including phase I biotransformation products such as α-zearalenol, β-zearalenol, α-zearalanol, β-zearalanol, and zearalanon, as well as phase II biotransformation products conjugated with glucose, sulfate, and glucuronic acid. The presence of ZEN in the food chain can pose a health threat to humans and livestock [3]. This promiscuous compound induces the expression of estrogen receptors [4], contributes to reproductive disorders, and can even lead to changes in reproductive organs. Zearalenone also disrupts the hormonal balance [2,5,6], metabolic profile [7], and gut microbiota [8]. It exerts hepatotoxic [9,10], immunotoxic [11], hematotoxic [12], and genotoxic effects [13].

In humans and animals, ingested ZEN initially interacts with the gastrointestinal system [14]. Two research problems can be formulated in this stage. Firstly, ZEN’s influence on intestinal health [15,16] and its carry-over from the intestinal digesta to intestinal tissues [16] has attracted considerable research interest in recent years. Secondly, in line with the hormesis paradigm [17], efforts are being made to evaluate the macroorganism’s response to low mycotoxin doses that are frequently encountered in feed materials. Researchers have also attempted to determine tissue and cell dysfunctions in mammals exposed to the pure parent compound without metabolites or modified mycotoxins [4,18]. In view of our previous findings [5,7,8,13], potential clinical symptoms of ZEN mycotoxicosis of varying severity were taken into account. Therefore, three ZEN doses were considered in the interpretation of our previous results and the findings of other authors: the lowest adverse effect observed level (LOAEL, >10 μg ZEN/kg BW) [19,20,21] that causes clinical symptoms [22]; (ii) the no adverse effect observed level (NOAEL, 10 μg ZEN/kg BW) as the highest dose that does not cause clinical symptoms (sub-clinical states) [23]; and (iii) the minimal anticipated biological effect level (MABEL, <10 ZEN/kg BW) as the lowest dose that can be measured in tissues, entering into positive interactions with the host at various stages in life [24,25].

The observation that mycotoxins modulate the expression of enzymes that participate in phase I and II biotransformation processes [26] was taken into consideration in the current experiment. The biotransformation of undesirable substances, including ZEN and its metabolites, is affected by two types of enzymes in cells. During phase I biotransformation, enzymes such as CYP P450 trigger chemical reactions (hydroxylation) to remove undesirable substances that can act as enzyme substrates [27]. These enzymes exhibit maximum activity in the liver and the small intestine, which are responsible for detoxication. They are also involved in the biosynthesis of steroid hormones [28] and fatty acid metabolism, as well as the activation/inactivation of drugs and other exogenous compounds in the body [29,30]. During phase II biotransformation, metabolic rates can be measured based on the activity of glutathione S-transferase (GST). The rate of biotransformation and the rate at which undesirable feed-borne substances are excreted from the body affect the response of the intestinal mucosal immune system to toxins. The π isoform of glutathione transferase (GSTπ1) codes for various proteins involved in these processes. Glutathione S-transferase reduces the biological activity of various (not only exogenous) substances by creating conjugates of glutathione with electrophilic drugs and other exogenous toxins. These conjugates protect the body against the adverse effects of oxidative stress and prevent damage to lipids and nucleic acids. It is worth noting that conjugates can break down again in the lumen of the gut, thus increasing the concentration of mycotoxins and theoretically increasing the activity of enzymes such as CYP1A1 and GSTπ1 [26,30].

The objective of this study was to determine how in vivo or low doses of ZEN (MABEL (5 µg/kg BW), the greatest values of NOAEL (10 µg/kg BW), and LOAEL (15 µg/kg BW)) administered orally for 42 days influence the level of ZEN and selected ZELs in intestinal wall tissues on different dates of exposure and contribute to the expression of genes encoding selected enzymes (CYP1A1 and GSTπ1) in the colon in prepubertal gilts.

## 2. Results

The present results confirmed that ZEN is biotransformed in the initial stage of the carry-over process when this mycotoxin and its metabolites are transported from the lumen of the gut to the intestinal wall. Parent mycotoxin and its two most widely investigated metabolites (α-ZEL and β-ZEL) are accumulated in this stage.

The above observation was confirmed by the mean concentrations of ZEN and its metabolites in the intestinal wall on different exposure days (see Figure 1). The percentage share of each substance in the total pool was consistent with the anticipated values for pigs, but the carry-over factor (CF) was much lower, which cannot be attributed solely to low-dose ZEN mycotoxicosis.

### 2.1. Experimental Feed

The analyzed feed did not contain mycotoxins, or its mycotoxin content was below the sensitivity of the method (VBS). The concentrations of masked mycotoxins were not analyzed.

### 2.2. Clinical Observations

The results presented in this paper were acquired during a large-scale experiment where clinical signs of ZEN mycotoxicosis were not observed. However, changes in specific tissues or cells were frequently noted in analyses of selected serum biochemical parameters, hematological parameters, the heart muscle, the bone marrow microenvironment, coronary artery, the genotoxicity of cecal water, selected steroid concentrations, gut microbiota parameters, and weight gains. Samples for laboratory analyses were collected from the same prepubertal gilts. The results of these analyses were presented in our previous studies [5,7,8,12,13,31,32].

### 2.3. Concentrations of Zearalenone and Its Metabolites in the Intestinal Tract of Prepubertal Gilts

Significant differences in ZEN levels between exposure date D1 and exposure dates D2 and D3 (see Table 1) were observed in the ascending colon centrifugal gyrus and the ascending colon centripetal gyrus in group ZEN5. In group ZEN10, significant differences were found in the transverse colon between D3 vs. D1 and D2. In group ZEN15, no significant differences were noted between exposure dates.

A comparison of ZEN concentrations in the examined segments of the intestinal tract (see Table 1) on the analyzed exposure dates in group ZEN5 revealed that: (i) in nearly all intestinal segments (excluding the ileum), ZEN concentrations were highest on D1; (ii) ZEN concentrations were highest in both gyri of the ascending colon, the jejunum, and cecum; (iii) ZEN concentrations were low and similar on all exposure dates only in the ileum (3.13, 3.26, and 4.10 ng/g, respectively); (iv) interestingly, ZEN concentrations tended to decrease over time in all intestinal segments.

The concentrations of ZEN differed in the ZEN10 group (see Table 1). In the duodenum, ZEN concentrations were very high and similar on all exposure dates (D1–D3) (64.36, 46.41, and 39.84 ng/g, respectively). In the remaining intestinal segments, ZEN concentrations were very low (3–20 ng/g) on D2 and D3, with a rising trend on successive dates of exposure.

In group ZEN15, ZEN concentrations were relatively high on D1 compared with the remaining dates of exposure (see Table 1). On D2, ZEN concentrations decreased in almost all sections of the intestine (excluding the ascending colon centrifugal gyrus—35.15 ng/g). On the last exposure date, ZEN concentrations increased in nearly all intestinal segments (excluding the centripetal gyri of the ascending colon—8.17 ng/g). The observed changes in ZEN values were relatively low in all intestinal segments, excluding the duodenum and the transverse colon.

In group ZEN5, significant differences in αZEL concentrations (see Table 2) were observed between D3 and D2 and between D2 and D1. In both cases, these differences were noted in the cecum and the centrifugal gyri of the ascending colon. In group ZEN10, significant differences were found in the cecum between D3 and D1, between D3 and D2, and in the centripetal gyri of the ascending colon between D3 and D1. In group ZEN 15, significant differences were noted in the duodenum, jejunum, cecum, and the centrifugal gyri of the ascending colon between D3 vs. D2 and D1 and between D2 and D1.

On D1 (see Table 2), significant differences in αZEL concentrations between groups ZEN10 and ZEN15 vs. group ZEN5 were found only in the cecum. On D2, the analyzed parameter differed significantly between group ZEN5 and group ZEN15 only in the centripetal gyri of the ascending colon and the transverse colon. On D3, significant differences between group ZEN5 and group ZEN15 were noted in the jejunum, cecum, and the centrifugal gyri of the ascending colon.

An analysis of αZEL concentrations (see Table 2) in different intestinal segments revealed that in group ZEN5, they were very low and similar (0.78 to 3.98 ng/g) on D1 and D3, whereas a wider range of values (1.07 ng/g on D1 to 6.69 ng/g on D3) was observed only in the ileum. On D2, αZEL levels were two or even three times higher than on D1 and D3, ranging from 7.00 ng/g in the descending colon to 49.47 ng/g in the cecum.

In group ZEN10, αZEL concentrations (see Table 2) were proportional to the time of exposure. However, on D2, αZEL concentrations in the ileum were below the values noted on D1 (difference of 2.00 ng/g) and D3 (difference of 4.03 ng/g).

In group ZEN15, αZEL concentrations were proportional in each intestinal segment. However, the analyzed parameter decreased in some segments (ileum) and increased rapidly in other parts of the intestinal tract (cecum). Alpha-ZEL levels were very low on exposure day 42, e.g., in the descending colon.

It should be noted that significant differences in *β*-ZEL concentrations were very rarely observed on exposure dates D1 and D2, and the highest number of significant differences was noted on D3 (see Table 3), in particular in group ZEN15. In group ZEN5, highly significant differences in *β*-ZEL concentrations were found only between D3 and D1 in the duodenum, jejunum, and cecum, and between D3 and D2 in the duodenum and jejunum. In group ZEN10, significant differences in the analyzed parameter were observed between D3 vs. D1 and D2 in the centrifugal gyri of the ascending colon. In group ZEN15, significant differences were noted between D3 and D1 in nearly all intestinal segments, excluding the ileum and the centripetal gyri of the ascending colon; significant differences were also noted between D3 and D2 in the jejunum, centripetal gyri of the ascending colon, transverse colon, and descending colon. Additionally, significant differences in *β*-ZEL levels were noted between D2 and D1 in the duodenum, jejunum, and cecum.

An analysis of *β*-ZEL concentrations (see Table 3) on different exposure dates revealed significant differences on D1 between group ZEN5 vs. groups ZEN10 and ZEN15 in the cecum and between group ZEN10 and group ZEN15 in the cecum. On D2, significant differences in *β*-ZEL concentrations were observed between group ZEN5 vs. groups ZEN10 and ZEN15 in the jejunum and cross member and between group ZEN10 and group ZEN15 in the jejunum. On D3, differences in the studied parameter were noted between group ZEN5 vs. groups ZEN10 and ZEN15 in the centrifugal gyri of the ascending colon and between group ZEN5 and group ZEN15 in the duodenum and descending colon.

### 2.4. Carry-Over Factor (CF)

Natural contamination of cereals and feeds with mycotoxins is frequently reported. Mycotoxins ingested by livestock with feed are carried over to food products of animal origin [16]. Undesirable substances are transferred from contaminated feed to animal tissues. The CF and the resulting health risks remain insufficiently investigated in vivo, in particular during low-dose mycotoxicosis, which is why they were analyzed in this study.

The carry-over rate of ZEN from the intestinal lumen to the intestinal wall was affected by the dose and time of exposure in each group (see Table 1). The highest CF values were noted on D1 in groups ZEN5 (centrifugal gyri of the ascending colon) and ZEN10 (duodenum). In group ZEN15, the highest CF values were observed in six of the eight investigated intestinal segments (excluding the jejunum and ileum). In turn, the lowest CF values were noted on D3 in all groups: in the transverse colon in group ZEN5 (8 × 10^−6^), in the jejunum in group ZEN10 (6 × 10^−6^), and in the jejunum in group ZEN15 (3 × 10^−6^).

The CF of αZEL in the intestinal tract ranged from 10^−4^ to 10^−6^ (see Table 2). The lowest values of this parameter (10^−6^) were noted in group ZEN5 on D3 in the transverse colon and the descending colon; in group ZEN10 on D1 in the descending colon and on D3 in the transverse colon and the descending colon; in group ZEN15 on D1 in the ileum, centripetal gyri of the ascending colon, transverse colon, and descending colon; on D2 in all intestinal segments, excluding the cecum; and on D3 in the ileum, transverse colon, and descending colon. Interestingly, CF values were lowest in the descending colon on all days of exposure. In turn, the highest values of CF (10^−4^) were found in group ZEN5 on D2 in the cecum; in group ZEN10 on D2 and D3 in the jejunum and cecum; and in group ZEN15 on D2 in the cecum. In 80% of the cases, the highest CF values were noted in the cecum.

The carry-over rates of β-ZEL (see Table 3) in different segments of the intestinal tract were determined within a narrower range of values (10^−5^ to 10^−8^) than the carry-over rates of ZEN and α-ZEL. The highest CF values were noted on D1 in group ZEN10 in the duodenum, jejunum, and centripetal gyri of the ascending colon; on D2 in group ZEN10 in the jejunum and cecum; and on D3 in group ZEN5 in the duodenum and jejunum. In group ZEN15, the highest CF values were observed on D3 in the ileum. The CF factor was calculated in 72 cases, and it reached 10^−8^ in 1 case (ileum, group ZEN15, D1), 10^−7^ in 17 cases, 10^−6^ in 50 cases, 10^−5^ in 3 cases, and 0 in 1 case (ileum, group ZEN5, D1).

### 2.5. Expression of CYP1A1 and GSTP1 Genes

Cytochrome P450 1A1 (CYP1A1) participates in the phase I metabolism of xenobiotics and drugs. It can exert protective effects on DNA and does not contribute to potentially cancerogenic DNA modifications, which could be attributed to the fact that CYP1A1 is highly active in the intestinal mucosa and prevents xenobiotics and carcinogens from reaching the circulatory system [33]. In this experiment, the *CYP1A1 m*RNA gene was generally silenced in both segments of the large intestine in all groups (see Figure 2). In the ascending colon, on D1 and D3, *CYP1A1 m*RNA expression was most strongly silenced in group C in comparison with the experimental groups. Similar results were reported by Sun et al. [6] in Leydig cells exposed in vitro to ZEN. In the ascending colon, on D1 and D3, *CYP1A1 m*RNA expression was most strongly silenced in group C (1.226 and 0.623, respectively). In contrast, on D2, *CYP1A1 m*RNA expression was more silenced in the experimental groups (1.103, 0.983, and 0.929, respectively) than in group C (1.313). The reverse was noted in the descending colon, where *CYP1A1 m*RNA expression was more silenced in the experimental groups on D1 (0.952, 0.613, and 0.676, respectively) and D3 (0.542, 0.480, and 0.462, respectively). On D2, *CYP1A1 m*RNA expression was most strongly silenced in groups C and ZEN5 (0.536 and 0.588, respectively).

Glutathione S-transferases (GSTs) are detoxification enzymes that catalyze conjugation reactions between endogenous glutathione and electrophilic metabolites produced during phase I biotransformation [34]. These enzymes protect cells against the harmful effects of electrophilic chemical compounds and oxidation products. Glutathione S-transferases are a family of dimeric enzymes responsible for the conjugation of exogenous and endogenous compounds with glutathione. Glutathione protects DNA against damage by binding toxic compounds in the cytoplasm and preventing them from interacting with nucleic acid [35].

In the present study, the expression of the GSTπ1 gene was silenced [34,36] as a result of low-dose zearalenone mycotoxicosis (see Figure 3). The suppression of gene expression was directly proportional to the zearalenone dose in the experimental groups. The expression of GSTP1 mRNA in both intestinal segments was highest in group C on all dates of exposure. In the ascending colon, significant differences were found between group C vs. groups ZEN10 and ZEN15 on all exposure dates (difference of 0.187 and 0.404 on D1; difference of 0.302 and 0.405 on D2; difference of 0.331 and 0.351 on D3, respectively). In group ZEN5, GSTP1 mRNA expression was also more strongly silenced than in group C, but the difference was not statistically notable. In the descending colon, notable differences were not observed on D1 and D2, whereas on D3, a significant difference (0.303) was noted between group C and group ZEN15.

## 3. Discussion

This study demonstrated that the absorption and biotransformation of ZEN and its metabolites in prepubertal gilts were highly individualized (see Table 1, Table 2 and Table 3). A comparison between the current and previous studies indicates that long-term exposure to low ZEN doses stimulates proliferation processes and migration on all dates of exposure [5,7,8,12,13,20,31,37]. The question that remains to be answered is why ZEN metabolites were detected in intestinal tissues if their concentrations were considerably lower in the blood [5] and the heart muscle [31] and if they were completely absent in the bone marrow microenvironment [12].

### 3.1. Zearalenone and its Metabolites

#### 3.1.1. Zearalenone Concentrations

An analysis of ZEN concentrations in gut tissue (see Table 1) revealed the highest values in the duodenum on exposure date D1 (ZEN levels were not always proportional to the ingested dose) in group ZEN10. This observation can be explained by the fact that in the initial stages of intestinal absorption, ZEN is most effectively absorbed in the duodenum, where it is hydroxylated to α-ZEL and β-ZEL [3]. This is because the duodenum and jejunum have firmly adherent mucus and polysaccharide layers [38], and prepubertal gilts have high requirements for estrogen and estrogenic compounds [5] due to supraphysiological hormonal levels [39]. In groups ZEN10 and ZEN15, higher ZEN concentrations could also be attributed to the presence of “free ZEN”, which contributes to: (i) the inhibition of steroidogenesis [6]; (ii) the conversion of testosterone (ZEN suppresses testosterone levels) to estradiol (which inhibits maturation processes in prepubertal gilts, [40]); and (iii) increased feed intake and the accumulation of energy [7], supporting homeostasis and, at later procreation of the organism, reproduction in prepubertal gilts [41]. It should also be noted that the period of adaptation to an ongoing mycotoxicosis ends after seven dates of exposure (i.e., on D1) [42].

On D2 and D3, ZEN levels in intestinal tissues increased (see Figure 1) in all groups, but values recorded in groups ZEN10 and ZEN15 (excluding the centrifugal gyri of the ascending colon) were proportionally lower than in group ZEN5. This is because the intestinal tract constitutes the first line of defense against the harmful effects of mycotoxins [14,43]. The intestines are covered by a layer of mucus and polysaccharides that adhere loosely or firmly to the intestinal wall [38] and determine the rate of absorption processes. Mycotoxins, including ZEN, exert negative effects mainly during prolonged exposure to high doses [14]. Zearalenone stimulated the proliferation of colonic cells in vitro [3]. Therefore, intestinal responses are determined mainly by the ZEN dose. Low doses promote proliferation and migration [44], while high doses have cytotoxic effects and inhibit biotransformation processes [3], which probably occurred in group ZEN15 in the early stages of exposure.

In view of the above, the present results suggest that the administered ZEN doses (LOAEL, NOAEL, and MABEL) were appropriate to reliably assess the carry-over of ZEN in the digestive tract of prepubertal gilts.

#### 3.1.2. Concentrations of ZEN and Its Metabolites

The levels of ZEN metabolites (α-ZEL and β-ZEL; see Table 2 and Table 3, Figure 1) in intestinal tissues were proportional to the administered doses and time of exposure. The proportions of α-ZEL and β-ZEL were similar to those noted in other studies [3], which is natural in our opinion [5,45]. The bioavailability of ZEN metabolites in intestinal tissues was influenced by biotransformation processes during pubescence. The distribution of concentration values in intestinal tissues preceded the values obtained in blood [5]. Zearalenone metabolites were not found in the blood of ZEN5 animals on D1 due to a very low supply of endogenous steroid hormones [5] (supraphysiological hormonal levels [39]). Mycoestrogens supplementation [46] can modify the levels of estrogen hormones [7,37] and the anti-Müllerian hormone [47].

The following arguments should be also considered: (i) adaptive processes end on D1 [42]; and (ii) ZEN metabolites can be used as substrates that regulate the expression levels of genes encoding hydroxysteroid dehydrogenases [20] that act as molecular switches and allow for the modulation of steroid hormone pre-receptors. In this study, these processes were particularly noticeable in group ZEN5, where ZEN concentrations decreased and α-ZEL concentrations increased on successive dates of exposure (see Table 2). The above indicates that in prepubertal gilts, even the smallest amounts of estrogenic compounds are used by the body [48] to make up for the deficiency of endogenous estrogen [5]. In the remaining groups, metabolite levels increased proportionally to the ingested ZEN dose.

Based on previous research findings [49,50] and extrapolation, the above observations could be attributed to the activity of transport proteins in the intestinal wall. The activity of the antiporter plays a very important role in the initial stage stages of mycotoxin biotransformation. Processes with the participation of antiport proteins are influenced by the availability of energy compounds (which were depleted) [7], which affect active ion pumps that transport ZEN from and to the intestinal lumen, thus stabilizing ZEN concentrations inside cells [51]. In enterocytes, numerous detoxification enzymes are localized near the cell wall. When mycotoxins and their metabolites do not undergo further biotransformation, they reach the cytosol and, consequently, the circulatory system. To prevent this from happening, active ion pumps remove mycotoxins and their metabolites from cells. These toxins return to the intestinal lumen and are subsequently recirculated to enterocytes. The described mechanisms restore energy reserves in cells, and mycotoxins can be metabolized again before they reach the cytosol and cause pathological states. Antiporter activity in the intestines is also determined by phase I enzymes, including the cytochrome P450 3A4 isoenzyme, which plays a key role in detoxification [15] and energy supply [7].

These observations could suggest that α-ZEL contributes to the modulation of life processes by participating in feminization processes in prepubertal gilts.

#### 3.1.3. Carry-Over Factor

One of the main objectives of risk management in the food processing industry is to protect public health through the effective identification and control of known threats and selecting the most appropriate strategies for mitigating these risks. The significance of the CF has to be understood before a given strategy is selected and implemented [1]. The CF is the concentration ratio of the undesirable substances (mycotoxin) in contaminated digesta to the concentrations of ZEN and its metabolites in gut tissues, which marks the beginning of biotransformation processes. Our previous research demonstrated that even MABEL doses can induce specific changes in homeostasis [52], metabolic processes [7], endocrine processes [5,6], and gut microbiota [8] in prepubertal gilts.

The CF values noted in the current study suggest that the accumulation of the parent compound in the intestinal wall was low and stable on the first two dates of exposure (D1 and D2). A different trend was noted in group ZEN10 in the duodenum, where ZEN is most easily and rapidly absorbed [38]. These values were sufficient to induce considerable disruptions in metabolic processes [18] and, most importantly, steroidogenesis [6,20,53,54] via estrogen receptors [4,52]. These findings confirm that the effects exerted by mycotoxins applied at low doses have not been fully elucidated in prepubertal females. However, on D3, the CF values were higher in the ZEN15 group than in the other groups, which may suggest the rate of biotransformation processes decreases over time. These observations are difficult to interpret because little is known about the rate and course of biological processes in sexually immature pigs. It can only be speculated that estrogen supply and estrogen demand reach equilibrium and that the steroid hormone profile is modified when testosterone is converted to estradiol [5,42] or when detoxification processes begin to dominate [55].

The CF values of both ZEN metabolites were characterized by different trends (Table 2 and Table 3). In general, this parameter was highest in group ZEN10 on all exposure dates, with a decreasing trend in the colon. On D3, the CF of α-ZEL in the ZEN10 group increased in all sections of the small intestine. The CF of β-ZEL was highest in group ZEN5 and also in the small intestine. Interestingly, a comparison of the CF values of both ZEN metabolites on all exposure dates revealed average values on D3. These observations suggest that after 21 days of exposure to the pure parent compound, life processes in prepubertal gilts are stabilized, and biotransformation processes are shifted towards detoxification [55,56]. This indicates that the macroorganism has developed tolerance to low ZEN doses due to the availability of “free ZEN” [20], its suppressive effects on testosterone and progesterone levels [6], and the negative feedback effect on FSH synthesis and secretion [5,57]. These effects could be ascribed to the fact that prepubertal gilts were fed a balanced diet [7,58], and the amount of gut microbiota was higher in the distal part of the intestinal tract [8], where mycotoxin concentrations were lower.

Contrary to the observations made in vitro by Alvarez-Ortega et al. [59], the present study did not provide any evidence to prove that exposure to low doses of ZEN, the determined concentrations of ZEN and its metabolites, and CF values enhanced proliferation processes in the analyzed tissues. It can be concluded that the noted CF values indicate that the tested ZEN doses did not exert harmful effects and that the macroorganism relatively easily adapted to these doses.

### 3.2. Expression of CYP1A1 and GSTπ1 Genes

The biotransformation of xenobiotics, including ZEN, often alters the biological activity and chemical properties of these compounds. Phase I metabolic processes usually enhance the functionality of their molecules, while phase II is often considered the detoxification stage. However, the nature and extent of these processes are highly compound-specific and can vary between individuals [2]. During the biotransformation process, ZEN is not only eliminated from the body, but the parent compound is inactivated, and the products of phase I biotransformation are activated. Various types of enzymes participate in these processes, including cytochrome P450 isoenzymes and CYP1A family enzymes in phase I, in particular in intestinal tissues. Microsomal GST enzymes are involved in successive phases of biotransformation [3].

The cytochrome P450 1A1 enzyme adds hydroxyl groups in the phase I metabolism of xenobiotics. Hydroxyl groups increase the polarity of xenobiotics, thus facilitating their excretion via urine, and they also act as sites for further modification in phase II biotransformation processes. As a result of enzymatic modifications, ZEN is partly transformed to more active forms (such as α-ZEL; it should be noted that pure ZEN was administered per os in this study) [2] that more easily interact with the molecular targets in cells (such as estrogen receptors) [60] and effectively participate in phase II biotransformation. These processes occur in two mutually opposing directions.

Analysis of the expression of the CYP1A1 gene in the ascending and descending colon (Figure 2) on dates D1 and D2 showed that the absorption and accumulation of ZEN and its metabolites in the gut tissues of prepubertal gilts (Figure 1) corresponded to the CF values in the analyzed intestinal segments (see Table 1). The above could be attributed to physiological estrogen deficiency [6,39] as well as the fact that the parent compound (ZEN) absorbed in intestinal tissues was biotransformed to a more active form (α-ZEL). These processes promote effective modulation of molecular targets in cells, facilitate steroidogenesis [5], and induce cross-talk between the receptors activated by undesirable substances (the estrogen receptor and the aryl hydrocarbon receptor). The interactions between these receptors increase the expression of the respective target genes [60].

It should also be noted that various substrates can stimulate or inhibit the activity of proteins, which significantly influences the biotransformation of ZEN catalyzed by these enzymes. In addition to protein polymorphism, the induction of biotransforming enzymes is also responsible for differences in the susceptibility to ZEN and the mycotoxin’s impact on gilts before puberty. Mean concentrations of ZEN and its metabolites in the intestinal wall (Figure 1) were inversely related to the expression of the CYP1A1 gene in two sections of the colon (see Figure 2), and these differences were more pronounced on successive dates of exposure [6]. The above can be attributed to the increasing accumulation of mycotoxins in the intestinal wall on day 42 of exposure (D3). In an in silico study, all mycotoxins (ZEN, α-ZEL, and β-ZEL) acted as substrates, inducers, and inhibitors ranging from 60% to 90%, 21% to 38%, and 23% to 32%, respectively, for the CYP1A1 isoform [15], which was not confirmed in vivo in this study. These observations could suggest that these mycotoxins suppress the expression of the CYP1A1 gene.

Glutathione S-transferases (GSTs) are a family of dimeric enzymes that conjugate exogenous and endogenous substances with glutathione. Glutathione prevents DNA damage by binding toxic compounds in the cytoplasm and preventing their interactions with nucleic acid. It should be noted that GSTs are a part of the unified cellular defense system [61]. In the GST family, the GSTπ1 subclass predominates in the colon [62]. The substrates for GSTs, in particular for the GSTπ1 subclass, include active metabolites of cyclophosphamide, platinum derivatives, and xenobiotics. GSTπ1 plays a special role in the conjugation of reactive cyclophosphamide metabolites with glutathione [63]. In the present study, a minor increase (non-significant) in GSTπ1 expression in the descending colon was noted only on D2 in groups ZEN10 and ZEN15 (Figure 3), which suggests that proliferation processes are somewhat dominant in enterocytes [64,65]. These results are difficult to interpret due to the scarcity of published data for comparisons. In most experimental weeks, GSTπ1 expression tended to be higher in the ascending colon and lower in the descending colon. Increased GSTπ1 expression in the ascending colon is attributable to hyperestrogenism in prepubertal gilts. Lower levels of GSTπ1 expression in the descending colon were also reported by Hokaiwado et al. [66], who concluded that GSTπ1 silencing decreases cell proliferation, promotes apoptosis, and contributes to controlled proliferation. According to other authors, GSTπ1 can be silenced in response to chemical stress [28,34], including exposure to ZEN. However, there are no published data that can be directly compared with our findings. Our previous studies revealed that both ZEN and deoxynivalenol (DON) could silence GSTπ1 expression [4,36].

As a result, long-term exposure to ZEN applied at low doses induced functional changes in the distal gastrointestinal tract in gilts before puberty. However, GSTπ1 expression in the descending colon was higher in group C on D2. This observation could be attributed to: (i) the balanced supply and demand (homeostasis) of GSTπ1 for the maintenance of cytoprotective and detoxification functions in phase II biotransformation processes; (ii) excess intracellular levels of GSTπ1 in the ascending and descending colon in response to high GSTπ1 expression [63]; or (iii) intensified apoptosis [66].

Exposure to low mycotoxin doses led to controlled silencing of GSTπ1 expression, thus promoting a balance between intestinal cells, the degree of exposure to toxic compounds, and the detoxifying effect.

## 4. Summary and Conclusions

The gut tissues are the first line of defense against any substances that enter the body. A healthy intestinal barrier guarantees homeostasis in the body. On the basis of the presented research, it could be hypothesized that the obtained results confirm the correctness of the adopted nomenclature of doses (LOAEL, NOAEL, and MABEL) used in the experience. Based on the observations made during 42 days of exposure to pure ZEN, it can be hypothesized that all mycotoxins (ZEN, α-ZEL, and β-ZEL) contribute to a balance between gut cells and the expression of genes that encode enzymes that participate in biotransformation processes in the large intestine, modulate feminization processes in prepubertal gilts, and elicit flexible, adaptive responses of the macroorganism to mycotoxin exposure at the analyzed doses.

## 5. Materials and Methods

### 5.1. General Information

All experimental procedures on animals were carried out in accordance with the Polish regulations specifying the conditions for conducting experiments on animals (Opinions No. 12/2016 and 45/2016/DLZ issued by the Local Ethics Committee for Animal Experimentation of the University of Warmia and Mazury in Olsztyn, Poland, on 30 November 2016). This article is a continuation of the previously published study protocol [32].

### 5.2. Experimental Feed

ZEN analytical samples were dissolved in 96 µl of 96% ethanol (SWW 2442-90, Polskie Odczynniki SA, Poland) in doses appropriate for various body weights (BWs). The food was placed in gel capsules saturated with the solution and kept at room temperature until the alcohol was evaporated. All groups of gilts received the same feed throughout the trial. The animals were weighed at weekly intervals, and the results were used to adjust the individual mycotoxin doses [8,13,67].

The feed given to all test animals was supplied by the same producer. The brittle feed was administered ad libitum twice a day at 8:00 a.m. and 5:00 p.m. throughout the experiment. The composition of the complete diet declared by the producer is presented in Table 4 [8,13,67]. Zearalenone analytical samples were dissolved in 96 µL of 96% ethanol (SWW 2442-90, Polskie Odczynniki SA, Poland) in appropriate weight doses. The feeds containing various amounts of ZEN in the alcoholic solution were placed in gel capsules. Prior to administration, the capsules were stored at room temperature until the alcohol evaporated. Pigs in the experimental groups received ZEN in gel capsules daily before morning feeding. The animals were weighed at weekly intervals to adjust the individual mycotoxin doses. The carrier was feed, and the gilts from group C received identical gel capsules without ZEN [7,8]. The feeds were supplied by the same manufacturer. During the experiment, the gilts were fed brittle fodder ad libitum twice a day (at 8:00 a.m. and 5:00 p.m.). The composition of the complete diets was specified by the producer, and it is presented in Table 4.

The proximate chemical composition of the diets fed to pigs in groups C, ZEN5, ZEN10, and ZEN15 was determined using the NIRS™ DS2500 F feed analyzer (FOSS, Hillerød, Denmark), a monochromator-based NIR reflectance and transflectance analyzer with a scanning range of 850–2500 nm [32].

#### Toxicological Analysis of Feed

Feed was analyzed for the presence of mycotoxins and their metabolites: ZEN, α-ZEL, and deoxynivalenol (DON). Mycotoxin concentrations in feed were determined by separation in immunoaffinity columns (Zearala-TestTM Zearalenone Testing System, G1012, VICAM, Watertown, MA, USA; DON-TestTM DON Testing System, VICAM, Watertown, MA, USA). Feed samples were ground in a laboratory mill. Ground samples of 25 g each were eluted with 150 mL of acetonitrile (90%) to extract the mycotoxins. A total of 10 mL of the resulting solution was withdrawn and diluted with 40 mL of water. The obtained solution (10 mL) was collected and passed through the immunoaffinity column (VICAM). The immunoaffinity bed in the column was subsequently washed with demineralized water (Millipore Water Purification System, Millipore S.A., Molsheim, France). The column was eluted with 99.8% methanol (LIChrosolvTM, No. 1.06 007, Merck-Hitachi, Germany) to remove the bound mycotoxin. The obtained solutions were analyzed by high-performance liquid chromatography (HPLC system, Hewlett Packard type 1050 and 1100), coupled with a diode array detector (DAD), a fluorescence detector (FLD), and chromatography columns (Atlantis T3 3 µm 3.0 150 mm Column No. 186003723, Waters, AN Etten-Leur, Ireland). Mycotoxins were separated in a mobile phase of acetonitrile:water:methanol (46:46:8, *v*/*v*/*v*). The flow rate was 0.4 mL/min. The limit of detection was set at 5 μg/kg of feed for DON and 2 μg/kg of feed for ZEN, based on validation of chromatographic methods for the determination of ZEN and DON levels in feed materials and feeds. Chromatographic methods were validated at the Department of Veterinary Prevention and Feed Hygiene [68]; see Appendix A.

### 5.3. Experimental Animals

An in vivo experiment involving 60 clinically healthy prepubertal gilts with initial BW of 14.5 ± 2 kg was performed at the Department of Veterinary Prevention and Feed Hygiene of the Faculty of Veterinary Medicine at the University of Warmia and Mazury in Olsztyn, Poland [7,32]. During the experiment, the animals were housed in pens, fed identical diets, and provided with ad libitum access to water. The gilts were randomly divided into a control group (group C; *n* = 15) and three experimental groups (ZEN5, ZEN10, and ZEN15; *n* = 15 each) [69,70]. Groups ZEN5, ZEN10, and ZEN15 were administered ZEN (Sigma-Aldrich Z2125-26MG, St. Louis, MO, USA) per os at 5 µg/kg BW, 10 µg/kg BW, and 15 µg/kg BW, respectively. Each experimental group was kept in a separate pen in the same building. Pens had an area of 25 m^2^ each, which is consistent with the applicable cross-compliance regulations (Regulation (EU) No 1306/2013 of the European Parliament and of the Council of 17 December 2013).

#### 5.3.1. Toxicological Analysis of Intestinal Tissues

##### Tissues Samples

Five prepubertal gilts from every group were euthanized on analytical date 1 (D1—exposure day 7), date 2 (D2—exposure day 21), and date 3 (D3—exposure day 42) by intravenous administration of pentobarbital sodium (Fatro, Ozzano Emilia BO, Italy) and bleeding. Immediately after cardiac arrest, tissue samples (approximately 1 *×* 1.5 cm) were collected from entire intestinal cross-sections, from the following segments of the gastrointestinal tract: duodenum—third part; jejunum and ileum—middle part; cecum—1 cm from the ileocecal valve; colon—middle part of the centrifugal gyri of the ascending colon and centripetal gyri of the ascending colon (ascending colon), transverse colon, and descending colon. The samples were rinsed with phosphate buffer and prepared for analyses. The collected samples were stored at a temperature of −20 °C.

##### Extraction Procedure

The presence of ZEN, α-ZEL, and β-ZEL in tissue samples was determined with the use of immunoaffinity columns. Tissue samples were transferred to centrifuge tubes and homogenized with 7 mL of methanol (99.8%) for 4 min. The tubes were vortexed 4 times at 5 min intervals, after which they were centrifuged at 5000 rpm for 15 min. Samples of 5 mL were collected from the suspension and combined with 20 mL of deionized water, and 12.5 mL of the resulting solution was used to extract ZEN. The supernatant was carefully collected and passed through immunoaffinity columns (Zearala-TestTM Zearalenone Testing System, G1012, VICAM, Watertown, MA, USA) at a rate of 1–2 drops per second. The immunoaffinity bed in the column was subsequently washed with demineralized water (Millipore Water Purification System, Millipore S.A., Molsheim, France). Isocratic elution was performed with 99.8% methanol (LIChrosolvTM, No. 1.06 007, Merck-Hitachi, Germany) to remove the bound mycotoxin. After extraction, the eluents were placed in a water bath at a temperature of 50 °C and were evaporated in a stream of nitrogen. Dry residues were stored at −20 °C until chromatographic analysis. Next, 0.5 mL of 99.8% acetonitrile (ACN) was added to dry residues to dissolve the mycotoxin. The process was monitored with the use of internal standards (Cayman Chemical 1180 East Ellsworth Road Ann Arbor, Michigan 48108 USA, ZEN-catalog number 11353; Batch 0593470-1; *a*-ZEN-catalog number 16549; Batch 0585633-2; *β*-ZEN-catalog number 19460; Batch 0604066-7).

##### Chromatographic Quantification of ZEN and Its Metabolites

Zearalenone and its metabolites were quantified at the Institute of Dairy Industry Innovation in Mrągowo, Poland. The biological activity of ZEN, α-ZEL, and β-ZEL in the bone marrow microenvironment was determined by combined separation methods involving immunoaffinity columns (Zearala-TestTM Zearalenone Testing System, G1012, VICAM, Watertown, MA, USA), Agilent 1260 liquid chromatography (LC) system, and a mass spectrometry system (MS, Agilent 6470). Samples were analyzed on a chromatographic column (Atlantis T3, 3 μm 3.0 × 150 mm, column No. 186003723, Waters, AN Etten-Leur, Ireland). The mobile phase was composed of 70% acetonitrile (LiChrosolvTM, No. 984730109, Merck-Hitachi, Mannheim, Germany), 20% methanol (LiChrosolvTM, No. 1.06 007, Merck-Hitachi, Mannheim, Germany), and 10% deionized water (MiliporeWater Purification System, Millipore S.A. Molsheim-France) with the addition of 2 mL of acetic acid per 1 L of the mixture. The flow rate was 0.4 mL/min., and the temperature of the oven column was 40 °C. The chromatographic analysis was completed in 4 min. The column was flushed with 99.8% methanol (LIChrosolvTM, No. 1.06 007, Merck-Hitachi, Mannheim, Germany) to remove the bound mycotoxin. The flow rate was 0.4 mL/min., and the temperature of the oven column was 40 °C. The chromatographic analysis was completed in 4 min.

Mycotoxin concentrations were determined with an external standard and expressed in ppb (ng/mL). Matrix-matched calibration standards were applied in the quantification process to eliminate matrix effects that can decrease sensitivity. Calibration standards were dissolved in matrix samples based on the procedure that was used to prepare the remaining samples. The material for calibration standards was free of mycotoxins. The limits of detection (LOD) for ZEN, α-ZEL, and β-ZEL were determined as the concentration at which the signal-to-noise ratio decreased to 3. The concentrations of ZEN, α-ZEL, and β-ZEL were determined in each group and on three analytical dates (see Table 1).

##### Mass Spectrometric Conditions

The mass spectrometer was operated with ESI in the negative ion mode. The MS/MS parameters were optimized for each compound. The linearity was tested by a calibration curve including six levels. Table 5 shows the optimized analysis conditions for the mycotoxins tested.

##### Carry-Over Factor

Carry-over toxicity occurs when the body is able to survive under the influence of low doses of mycotoxins. Mycotoxins may impair the functions of tissues or organs [72] and modify their biological activity [5,7]. CF was determined in the intestinal tissues when the daily dose of ZEN (5 µg ZEN/kg BW, 10 µg ZEN/kg BW or 15 µg ZEN/kg BW) administered to each animal was equivalent to 560–32251.5 µg ZEN/kg complete diet, depending on the daily feed intake. The concentrations of mycotoxins in the tissues were expressed as the dry matter content of the samples.

The CF was calculated as follows:CF = toxin concentration in tissue [ng/g]/toxin concentration in diet [ng/g]

##### Statistical Analysis

The data were statistically processed at the Department of Discrete Mathematics and Theoretical Computer Science, Faculty of Mathematics and Computer Science, University of Warmia and Mazury in Olsztyn. The bioavailability of ZEN and its metabolites in gut tissues was analyzed in group C and three experimental groups on three analytical dates. Results are expressed as means (±) with standard deviation (SD). The following parameters were analyzed: (i) the differences in the mean values for the various doses of ZEN (experimental groups) and the control group on the three analytical dates, and (ii) the differences in the mean values for the individual doses (groups) of ZEN at the three dates. Differences between mean values were determined using one-way ANOVA. If there were significant differences between groups, the differences between the pairs of means were determined using Tukey’s multiple comparison test. If all values were below the LOD (mean and variance equal to zero) in either group, values in the remaining groups were analyzed by one-way ANOVA (if the number of remaining groups was greater than two), and the means of these groups were compared with zero on Student’s t-test. The differences between the groups were determined by Student’s t-test. The results were considered to be highly significant at *p* <0.01 (**) and significant at 0.01 < *p* <0.05 (*). The data were statistically processed using Statistica v.13 software (TIBCO Software Inc., Silicon Valley, CA, USA, 2017). Dose–response relationships were established using Pearson’s correlation analysis. Differences were considered significant at *p* ≤ 0.05. Results are presented as the means ± standard error of the mean (SEM).

### 5.4. Expression of CYP1A1 and GSTπ1

#### 5.4.1. Sampling and Storage for RNA Extraction

Immediately after cardiac arrest, tissue samples were collected from the mid-ascending and descending colon. The samples were stored in RNAlater (Sigma-Aldrich; Taufkirchen, Germany) according to the manufacturer’s instructions. Tissue samples were collected at the same time.

#### 5.4.2. Complete RNA Extraction and cDNA Synthesis

Total RNA was extracted from RNAlater-preserved tissues (approx. 20 mg per sample; *n* = 5 in each treatment group) using the Total RNA Mini isolation kit (A&A Biotechnology; Gdansk, Poland) according to the manufacturer’s protocol. RNA samples were incubated with RNase-free DNase I (Roche Diagnostics; Mannheim, Germany) to prevent contamination of genomic DNA. The overall RNA quality and purity of all samples were assessed with a BioPhotometer (Eppendorf; Hamburg, Germany), and the results were used for cDNA synthesis with the RevertAid ™ First Strand cDNA Synthesis Kit (Fermentas; Burlington, Canada). The cDNA synthesis reaction mixture for each sample contained 1 µg of total RNA and 0.5 µg of oligo (dT) primer, and the reaction was performed according to the manufacturer’s protocol. The first synthesized cDNA strand was stored at −20 °C until further analysis.

#### 5.4.3. qPCR

Real-time PCR primers for the CYP1A1 and GSTπ1 mRNAs were designed using the Primer-BLAST tool based on the reference species (Table 6). The real-time PCR test was performed on an ABI 7500 Real-time PCR system thermalcycler (Applied Biosystems; USA) in singleplex mode. Subsequent treatments were applied in accordance with the producents’ recommendations.

The quantitative cycle (Cq) values of qPCR were converted to copy number using a standard curve plot (Cq vs. log copy number) according to the methodology developed by [74] and described by Spachmo and Arukwe [75].

The rationale for the use of the standard curve is based on the assumption that the unknown samples have an equal amplification efficiency (usually above 90%), which is checked before extrapolating the unknown standards to the standard curve [75]. To generate standard curves, the purified PCR products of each mRNA were used to prepare a series of 6 10-fold dilutions with known copy number amounts that were used as templates in real-time PCR. The Cq values obtained for each dilution series were plotted against the log copy number and used to extrapolate the unknown samples to the copy number. The mRNA copy numbers of the samples collected from all experimental groups in each exposure date were divided by the averaged numbers from the C group, determined at the beginning of the experiment (control 0d), to obtain relative expression values, which were presented as the expression ratio (R).

#### 5.4.4. Statistical Analysis

The expression of the CYP1A1 and GSTP1 genes in the ascending and descending colon was presented as mean (±) SD values for each sample. The results were analyzed with Statistica software (StatSoft Inc., Tulsa, OK, USA). Mean values in the control and experimental groups were compared by a one-way ANOVA with repeated measures based on the dose of ZEN administered to the gilts before puberty. If differences between the groups were found, a post hoc Tukey test was performed to determine which pairs of mean groups were significantly different. In ANOVA, group samples were taken from normally distributed populations with the same variance. If the above assumptions were not met in all cases, the equality of the mean groups was tested using the Kruskal–Wallis rank test and the multiple-comparisons test in ANOVA. Different pairs of groups were identified by multiple post hoc comparisons of the rank means for all groups.

## Figures and Tables

**Figure 1 toxins-14-00354-f001:**
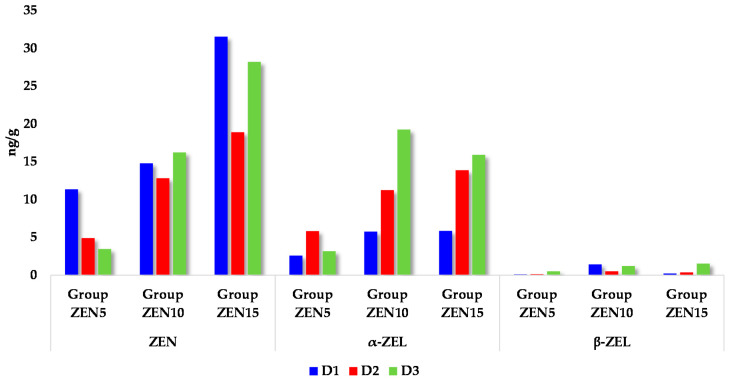
Mean values of ZEN and its metabolites (ng/g) in the intestinal wall in all experimental groups and on each exposure date. Key: D1—exposure day 7; D2—exposure day 21; D3—exposure day 42. Experimental groups: group ZEN5—5 µg ZEN/kg BW; group ZEN10—10 µg ZEN/kg BW; group ZEN15—15 µg ZEN/kg BW.

**Figure 2 toxins-14-00354-f002:**
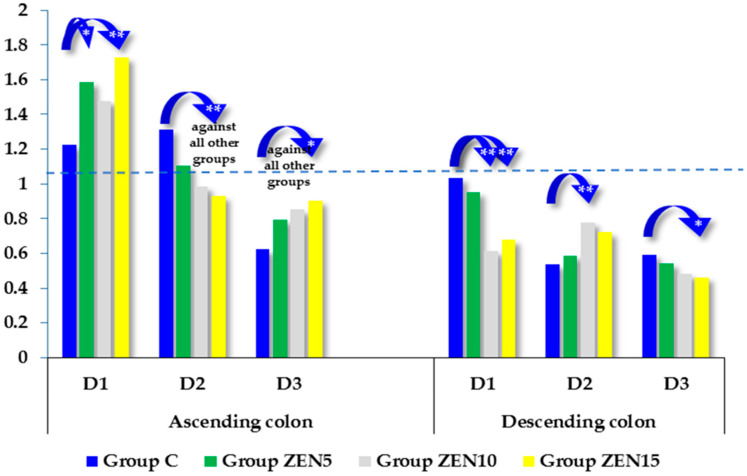
The expression of CYP1A1 in the ascending colon and descending colon at different dates of exposure. Bars represent the mean values (*n* = 5) of enzyme gene expression ratios (±) relative to the control sample at the beginning of the experiment (ER = 1.00; dashed line). Asterisks (*) denote experimental groups (group ZEN5—5 µg ZEN/kg BW; group ZEN10—10 µg ZEN/kg BW; group ZEN15—15 µg ZEN/kg BW), which differed significantly in mRNA levels compared to the corresponding control (C) groups (* *p* ≤ 0.05, ** *p* ≤ 0.01).

**Figure 3 toxins-14-00354-f003:**
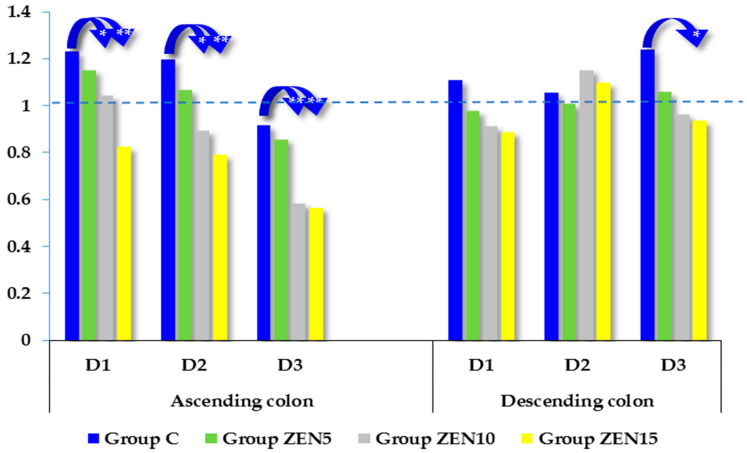
The expression of GSTπ1 in the ascending colon and descending colon at different dates of exposure. Bars represent the mean values (*n* = 5) of enzyme gene expression ratios (±) relative to the control sample at the beginning of the experiment (ER = 1.00; dashed line). Asterisks (*) denote experimental groups (group ZEN5—5 µg ZEN/kg BW; group ZEN10—10 µg ZEN/kg BW; group ZEN15—15 µg ZEN/kg BW), which differed significantly in mRNA levels compared to the corresponding control (C) groups (* *p* ≤ 0.05, ** *p* ≤ 0.01).

**Table 1 toxins-14-00354-t001:** The transfer coefficient and mean (±) concentration of ZEN (ng/g) in the intestines of gilts before puberty.

Exposure Dates	Feed Intake (kg/day)	Total ZEN Doses in Groups (µg/kg BW)	Tissues	Group ZEN5 (ng/g)	Carry-Over Factor	Group ZEN10 (ng/g)	Carry-Over Factor	Group ZEN15 (ng/g)	Carry-Over Factor
D1	0.8	80.5/161.9/242.7	Duodenum	12.81 ± 8.49	1 × 10^−4^	64.36 ± 71.16	3 × 10^−4^	48.35 ± 43.92	1 × 10^−4^
			Jejunum	15.59 ± 20.40	1 × 10^−4^	15.24 ± 13.65	9 × 10^−5^	13.31 ± 4.92	5 × 10^−5^
			Ileum	3.13 ± 0.69	3 × 10^−5^	4.67 ± 4.50	2 × 10^−5^	18.36 ± 13.81 ^c^	7 × 10^−5^
			Cecum	15.07 ± 8.14	1 × 10^−4^	6.84 ± 4.18	4 × 10^−5^	47.99 ± 34.30 ^d^	1 × 10^−4^
			CFG	14.74 ± 7.59	1 × 10^−4^	8.63 ± 5.09	5 × 10^−5^	33.13 ± 48.53	1 × 10^−4^
			CPG	18.24 ± 12.39	2 × 10^−4^	9.67 ± 4.95	5 × 10^−5^	28.48 ± 17.20	1 × 10^−4^
			Transverse colon	4.59 ± 2.74	5 × 10^−5^	4.30 ± 2.99	2 × 10^−5^	31.13 ± 20.40 ^c^^,^^d^	1 × 10^−4^
			Descending colon	6.11 ± 3.67	7 × 10^−5^	4.50 ± 2.68	2 × 10^−5^	31.69 ± 38.52	1 × 10^−4^
D2	1.1	101.01/196.9/298.2	Duodenum	7.67 ± 4.56	7 × 10^−5^	46.41 ± 33.57 ^c^	2 × 10^−4^	8.23 ± 10.24 ^d^	2 × 10^−5^
			Jejunum	6.81 ± 5.80	6 × 10^−5^	4.95 ± 1.94	2 × 10^−5^	5.56 ± 5.97	1 × 10^−5^
			Ileum	3.26 ± 2.32	3 × 10^−5^	3.30 ± 2.07	1 × 10^−5^	11.66 ± 5.46 ^cc,^ ^dd^	3 × 10^−5^
			Cecum	8.52 ± 6.13	8 × 10^−5^	12.56 ± 10.45	6 × 10^−5^	38.11 ± 38.07	1 × 10^−4^
			CFG	3.68 ± 1.06 ^aa^	3 × 10^−5^	12.13 ± 9.63	6 × 10^−5^	35.15 ± 37.92	1 × 10^−4^
			CPG	4.73 ± 1.76 ^a^	4 × 10^−5^	8.22 ± 8.53	4 × 10^−5^	24.82 ± 19.84	8 × 10^−5^
			Transverse colon	2.19 ± 1.31	2 × 10^−5^	7.64 ± 3.91	3 × 10^−5^	9.39 ± 8.98	3 × 10^−5^
			Descending colon	2.23 ± 1.18	2 × 10^−5^	7.31 ± 4.09	3 × 10^−5^	18.35 ± 13.56 ^c^	6 × 10^−5^
D3	1.6	128.3/481.4/716.7	Duodenum	8.84 ± 4.20	6 × 10^−5^	39.84 ± 2.22 ^cc^	8 × 10^−5^	57.34 ± 8.98 ^cc,^^dd^	8 × 10^−5^
			Jejunum	2.83 ± 4.26	2 × 10^−5^	3.02 ± 3.76	6 × 10^−6^	8.03 ± 0.98	1 × 10^−5^
			Ileum	4.10 ± 2.87	3 × 10^−5^	8.34 ± 0.94 ^c^	1 × 10^−5^	15.11 ± 0.66 ^cc,^^dd^	2 × 10^−5^
			Cecum	4.46 ± 4.73	3 × 10^−5^	17.17 ± 20.89	3 × 10^−5^	39.86 ± 2.30 ^cc^	5 × 10^−5^
			CFG	1.76 ± 2.07 ^aa^	1 × 10^−5^	16.73 ± 22.78	3 × 10^−5^	18.10 ± 2.50 ^cc,^ ^d^	6 × 10^−5^
			CPG	2.21 ± 1.56 ^a^	1 × 10^−5^	8.91 ± 2.80 ^cc^	1 × 10^−5^	18.17 ± 2.78 ^cc,^^dd^	2 × 10^−5^
			Transverse colon	1.10 ± 1.15	8 × 10^−6^	22.70 ± 12.87 ^a,^ ^b^^,^^c^	4 × 10^−5^	20.19 ± 1.30 ^c^	2 × 10^−5^
			Descending colon	2.30 ± 1.35	1 × 10^−5^	13.26 ± 16.42	2 × 10^−5^	18.94 ± 2.11	2 × 10^−5^

**Key**: CFG—Ascending colon centrifugal gyrus; CPG—Ascending colon centripetal gyrus; exposure dates: D1—exposure day 7; D2—exposure day 21; D3—exposure day 42. Experimental groups: group ZEN5—5 µg ZEN/kg BW; group ZEN10—10 µg ZEN/kg BW; group ZEN15—15 µg ZEN/kg BW. The statistically notable differences were determined at **^a^**, **^b^**, **^c^**, **^d^** *p* ≤ 0.05 and **^aa^**, **^cc^**, **^dd^** *p* ≤ 0.01; **^a^**, **^aa^** notable difference between exposure date D1 and exposure dates D2 and D3; **^b^** notable difference between exposure date D2 and exposure date D3; **^c^**, **^cc^** notable difference between group ZEN5 and groups ZEN10 and ZEN15; **^d^**, **^dd^** notable difference between group ZEN10 and group ZEN15.

**Table 2 toxins-14-00354-t002:** The transfer coefficient and mean (±) concentration of *α*ZEL (ng/g) in the intestines of gilts before puberty.

Exposure Date	Tissues	Group ZEN5 (ng/g)	Carry-Over Factor	Group ZEN10 (ng/g)	Carry-Over Factor	Group ZEN15 (ng/g)	Carry-Over Factor
D1	Duodenum	3.72 ± 1.68	4 × 10^−5^	6.25 ± 2.57	3 × 10^−5^	6.66 ± 6.32	2 × 10^−5^
	Jejunum	2.85 ± 2.78	3 × 10^−5^	5.24 ± 3.95	3 × 10^−5^	9.62 ± 1.66	3 × 10^−5^
	Ileum	1.07 ± 0.21	1 × 10^−5^	6.46 ± 7.25	3 × 10^−5^	2.88 ± 3.21	1 × 10^−5^
	Cecum	3.45 ± 1.05	4 × 10^−5^	12.01 ± 4.36 ^cc^	7 × 10^−5^	11.85 ± 2.04 ^cc^	4 × 10^−5^
	CFG	3.15 ± 0.49	3 × 10^−5^	5.58 ± 1.83	3 × 10^−5^	4.59 ± 2.78	1 × 10^−5^
	CPG	3.98 ± 2.43	4 × 10^−5^	6.88 ± 2.65	4 × 10^−5^	6.41 ± 4.58	2 × 10^−5^
	Transverse colon	1.07 ± 0.95	1 × 10^−5^	2.26 ± 1.74	1 × 10^−5^	2.05 ± 0.99	8 × 10^−6^
	Descending colon	1.48 ± 0.60	1 × 10^−5^	1.47 ± 0.80	9 × 10^−6^	2.68 ± 3.04	1 × 10^−5^
D2	Duodenum	6.43 ± 2.36	6 × 10^−5^	9.12 ± 8.11	4 × 10^−5^	6.97 ± 0.31	2 × 10^−5^
	Jejunum	5.36 ± 3.87	5 × 10^−5^	22.37 ± 20.81	1 × 10^−4^	14.28 ± 1.92 ^a^	4 × 10^−5^
	Ileum	6.58 ± 7.01	6 × 10^−5^	4.46 ± 2.99	2 × 10^−5^	8.49 ± 2.52	2 × 10^−5^
	Cecum	11.23 ± 4.47 ^aa^	1 × 10^−4^	22.74 ± 17.44	1 × 10^−4^	35.59 ± 14.60 ^a^	1 × 10^−4^
	CFG	6.01 ± 2.01 ^a^	5 × 10^−5^	10.80 ± 10.41	5 × 10^−5^	10.88 ± 1.74 ^aa^	3 × 10^−5^
	CPG	7.01 ± 2.76	6 × 10^−5^	11.86 ± 4.88	6 × 10^−5^	27.08 ± 15.25 ^c^	9 × 10^−5^
	Transverse colon	2.42 ± 1.59	2 × 10^−5^	5.86 ± 2.61	2 × 10^−5^	6.65 ± 1.55 ^c^	2 × 10^−5^
	Descending colon	1.51 ± 0.88	1 × 10^−5^	2.75 ± 1.74	1 × 10^−5^	1.07 ± 0.40	3 × 10^−6^
D3	Duodenum	3.72 ± 1.68	2 × 10^−5^	30.00 ± 22.55 ^c^	6 × 10^−5^	20.12 ± 4.65 ^aa,^ ^bb^	2 × 10^−5^
	Jejunum	2.85 ± 2.78	2 × 10^−5^	16.42 ± 5.54 ^cc^	3 × 10^−5^	19.64 ± 1.82 ^aa,^ ^b^^,^ ^cc^	2 × 10^−5^
	Ileum	6.69 ± 1.68	5 × 10^−5^	4.60 ± 4.00	9 × 10^−6^	7.32 ± 4.10	1 × 10^−5^
	Cecum	3.45 ± 1.05 ^bb^	2 × 10^−5^	49.47 ± 9.75 ^aa,^ ^bb^^,^ ^cc^	1 × 10^−4^	37.04 ± 5.08 ^a,^ ^cc^	5 × 10^−5^
	CFG	3.15 ± 0.49 ^b^	2 × 10^−5^	12.67 ± 8.59 ^c^	2 × 10^−5^	15.56 ± 1.34 ^aa,^ ^b^^,^ ^c^	2 × 10^−5^
	CPG	3.98 ± 2.43	3 × 10^−5^	25.84 ± 10.70 ^a,^ ^c^	5 × 10^−5^	16.13 ± 11.21	2 × 10^−5^
	Transverse colon	0.80 ± 0.94	6 × 10^−6^	8.08 ± 7.40	1 × 10^−5^	5.91 ± 6.88	8 × 10^−6^
	Descending colon	0.78 ± 0.74	6 × 10^−6^	7.00 ± 5.59	1 × 10^−5^	5.53 ± 3.86	4 × 10^−6^

**Key**: CFG—Ascending colon Centrifugal gyrus; CPG—Ascending colon Centripetal gyrus; Exposure dates: D1—exposure day 7; D2—exposure day 21; D3—exposure day 42. Experimental groups: group ZEN5—5 µg ZEN/kg BW; group ZEN10—10 µg ZEN/kg BW; group ZEN15—15 µg ZEN/kg BW. LOD > values below the limit of detection were expressed as 0. The statistically notable differences were determined at **^a^**, **^b^**, **^c^** *p* ≤ 0.05 and **^aa^**, **^bb^**, **^cc^** *p* ≤ 0.01; **^a^**, **^aa^** notable difference between exposure date D1 and exposure date D2 and D3; **^b^**, **^bb^** notable difference between exposure date D2 and exposure date D3; **^c^**, **^cc^** notable difference between group ZEN5 and groups ZEN10 and ZEN15.

**Table 3 toxins-14-00354-t003:** The transfer coefficient and mean (±) concentration of *β*ZEL (ng/g) in the intestines of gilts before puberty.

Exposure Date	Tissue	Group ZEN5 (ng/g)	Carry-Over Factor	Group ZEN10 (ng/g)	Carry-over Factor	Group ZEN15 (ng/g)	Carry-Over Factor
D1	Duodenum	0.16 ± 0.09	1 × 10^−6^	3.49 ± 6.14	2 × 10^−5^	0.29 ± 0.07	1 × 10^−6^
	Jejunum	0.07 ± 0.08	8 × 10^−7^	2.17 ± 3.61	1 × 10^−5^	0.12 ± 0.02	4 × 10^−7^
	Ileum	0	0	0.64 ± 0.77	3 × 10^−6^	0.02 ± 0.03	8 × 10^−8^
	Cecum	0.14 ± 0.04	1 × 10^−6^	1.19 ± 0.20 ^cc^	7 × 10^−6^	0.54 ± 0.10 ^cc,^ ^dd^	2 × 10^−6^
	CFG	0.13 ± 0.07	1 × 10^−6^	0.44 ± 0.28	2 × 10^−6^	0.30 ± 0.02	1 × 10^−6^
	CPG	0.18 ± 0.10	2 × 10^−6^	2.37 ± 3.66	1 × 10^−5^	0.27 ± 0.09	1 × 10^−6^
	Transverse colon	0.12 ± 0.18	1 × 10^−6^	0.77 ± 1.10	4 × 10^−6^	0.09 ± 0.06	3 × 10^−7^
	Descending colon	0.02 ± 0.01	2 × 10^−7^	0.22 ± 0.17	1 × 10^−6^	0.05 ± 0.01	2 × 10^−7^
D2	Duodenum	0.08 ± 0.10	7 × 10^−7^	0.26 ± 0.17	1 × 10^−6^	0.08 ± 0.09 ^aa^	2 × 10^−7^
	Jejunum	0.07 ± 0.10	6 × 10^−7^	1.11 ± 0.11 ^cc^	5 × 10^−6^	0.69 ± 0.11 ^aa,^ ^cc,^ ^dd^	2 × 10^−6^
	Ileum	0.14 ± 0.11	1 × 10^−6^	0.27 ± 0.18	1 × 10^−6^	0.05 ± 0.06	1 × 10^−7^
	Cecum	0.30 ± 0.19	2 × 10^−6^	1.07 ± 1.00	5 × 10^−6^	1.10 ± 0.26 ^a^	3 × 10^−6^
	CFG	0.17 ± 0.13	7 × 10^−7^	0.45 ± 0.29	2 × 10^−6^	0.24 ± 0.16	8 × 10^−7^
	CPG	0.13 ± 0.11	1 × 10^−6^	0.41 ± 0.12	2 × 10^−6^	0.46 ± 0.53	1 × 10^−6^
	Transverse colon	0.03 ± 0.03	1 × 10^−7^	0.27 ± 0.09 ^cc^	1 × 10^−6^	0.25 ± 0.05 ^cc^	8 × 10^−7^
	Descending colon	0.03 ± 0.02	1 × 10^−7^	0.23 ± 0.27	1 × 10^−6^	0.08 ± 0.04	2 × 10^−7^
D3	Duodenum	1.11 ± 0.67 ^aa,^ ^bb^	8 × 10^−6^	2.21 ± 0.73	5 × 10^−6^	3.06 ± 0.17 ^aa,^ ^cc^	4 × 10^−6^
	Jejunum	0.91 ± 0.45 ^aa,^ ^bb^	7 × 10^−6^	2.46 ± 1.51	5 × 10^−6^	3.08 ± 0.16 ^aa,^ ^bb^	4 × 10^−6^
	Ileum	0.21 ± 0.28	1 × 10^−6^	0.89 ± 1.32	1 × 10^−6^	0.61 ± 0.52	8 × 10^−7^
	Cecum	0.63 ± 0.24 ^aa^	4 × 10^−6^	1.18 ± 0.38	2 × 10^−6^	1.15 ± 0.19 ^a^	1 × 10^−6^
	CFG	0.48 ± 0.34	3 × 10^−6^	1.37 ± 0.18 ^a,^ ^b, cc^	2 × 10^−6^	1.66 ± 0.21 ^a,^ ^b, cc^	2 × 10^−6^
	CPG	0.47 ± 0.41	3 × 10^−6^	0.27 ± 0.20	5 × 10^−7^	0.80 ± 0.13	1 × 10^−6^
	Transverse colon	0.20 ± 0.14	1 × 10^−6^	0.76 ± 0.59	1 × 10^−6^	1.06 ± 0.20 ^aa,^ ^bb^	1 × 10^−6^
	Descending colon	0.16 ± 0.16	1 × 10^−6^	0.53 ± 0.19	1 × 10^−6^	0.79 ± 0.11 ^aa,^ ^bb, c^	1 × 10^−6^

**Key**: CFG—ascending colon centrifugal gyrus; CPG—ascending colon centripetal gyrus; exposure dates: D1—exposure day 7; D2—exposure day 21; D3—exposure day 42. Experimental groups: group ZEN5—5 µg ZEN/kg BW; group ZEN10—10 µg ZEN/kg BW; group ZEN15—15 µg ZEN/kg BW. LOD > values below the limit of detection were expressed as 0. The statistically notable differences were determined at **^a^**, **^b^**, **^c^** *p* ≤ 0.05 and **^aa^**, **^bb^**, **^cc^**, **^dd^** *p* ≤ 0.01; **^a^**, **^aa^** notable difference between exposure date D1 and exposure dates D2 and D3; **^b^**, **^bb^** notable difference between exposure date D2 and exposure date D3; **^c^**, **^cc^** notable difference between group ZEN5 and groups ZEN10 and ZEN15; **^dd^** notable difference between group ZEN10 and group ZEN15.

**Table 4 toxins-14-00354-t004:** Declared composition of the complete diet [12].

Parameters	Composition Declared by the Manufacturer (%)
Soybean meal	16
Wheat	55
Barley	22
Wheat bran	4.0
Chalk	0.3
Zitrosan	0.2
Vitamin–mineral premix ^1^	2.5

^1^ Composition of the vitamin-mineral premix per kg: vitamin A—500.000 IU; iron—5000 mg; vitamin D3—100.000 IU; zinc—5000 mg; vitamin E (alpha-tocopherol)—2000 mg; manganese—3000 mg; vitamin K—150 mg; copper (CuSO4·5H2O)—500 mg; vitamin B1—100 mg; cobalt—20 mg; vitamin B2—300 mg; iodine—40 mg; vitamin B6—150 mg; selenium—15 mg; vitamin B12—1500 μg; L-lysine—9.4 g; niacin—1200 mg; DL—methionine+cystine—3.7 g; pantothenic acid—600 mg; L-threonine—2.3 g; folic acid—50 mg; tryptophan—1.1 g; biotin—7500 μg; phytase+choline—10 g; ToyoCerin probiotic+calcium—250 g; antioxidant+mineral phosphorus and released phosphorus—60 g; magnesium—5 g; sodium and calcium—51 g.

**Table 5 toxins-14-00354-t005:** Optimized conditions for mycotoxins tested [71].

Analyte	Precursor	Quantification Ion	Confirmation Ion	LOD (ng mL^−1)^	LOQ (ng mL^−1)^	Linearity (%R^2^)
**ZEN**	317.1	273.3	187.1	0.03	0.1	0.999
**α-ZEL**	319.2	275.2	160.1	0.3	0.9	0.997
**β-ZEL**	319.2	275.2	160.1	0.3	1	0.993

**Table 6 toxins-14-00354-t006:** Real-time PCR primers for the proposed study.

Primer	Sequence (5′→3′)		Amplicon Length (bp)	References
**CYP1A1**	Forward	cagagccgcagcagccaccttg	226	[68]
	Reverse	ggctcttgcccaaggtcagcac		
**GSTP1**	Forward	acctgcttcggattcaccag	178	[68]
	Reverse	ctccagccacaaagccctta		
***β-*Actin**	Forward	catcaccatcggcaaaga	237	[73]
	Reverse	gcgtagaggtccttcctgatgt		

## Data Availability

The data presented in this study are available in this article and Appendix A.

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
