# Peer review of "Carry-Over of Zearalenone and Its Metabolites to Intestinal Tissues and the Expression of CYP1A1 and GSTπ1 in the Colon of Gilts before Puberty"

_toxins, 2022, doi:10.3390/toxins14050354_

Round 1
Reviewer 1 Report
Summary and general appreciation:
This study aimed to determine whether low doses of zearalenone (ZEN) influence 5 the carry-over of ZEN and its metabolites to intestinal tissues and the expression of CYP1A1 and 6 GSTπ1 in the large intestine. I think the is well written and structured and I think it is very important for the scientific community and falls within the scope of this journal.
My major concern relies in fact that no any validation results were presented for ZEN determination in tissues or feed. The authors refer the validation methodologies, but do not mention any validation data. I think the manuscript should be accepted if these results are to be provided, at least as supplementary material.
Minor comments:
Abstract/Key contribution
I don’t think the key contribution is evident enough in the abstract.
Author Response
This study aimed to determine whether low doses of zearalenone (ZEN) influence 5 the carry-over of ZEN and its metabolites to intestinal tissues and the expression of CYP1A1 and 6 GSTπ1 in the large intestine. I think the is well written and structured and I think it is very important for the scientific community and falls within the scope of this journal.
My major concern relies in fact that no any validation results were presented for ZEN determination in tissues or feed. The authors refer the validation methodologies, but do not mention any validation data. I think the manuscript should be accepted if these results are to be provided, at least as supplementary material.
Responding to the Reviewer's comments
- Thank you for your favorable review. At the same time, in response to the presented remark, we have introduced appropriate additions to the text (in red). For the validation of mycotoxins in feed, we submitted the validation data in the form of supplementary material (Table S1);
Minor comments:
Abstract/Key contribution
I don’t think the key contribution is evident enough in the abstract
- Abstract / Key contribution - we changed the content by introducing bolder and expressive statements that motivate our scientific activity

Reviewer 2 Report
Dear authors,
Congratulations on this excellent study and well written article based on an in vivo experiment with many results. The aim of this study was to determine whether low doses of zearalenone (ZEN) influence the carry-over of ZEN and its metabolites to intestinal tissues and the expression of CYP1A1 and GSTπ1 in the large intestine. My comments:
- In such an experiment, why do not you analyze the feed for other mycotoxins, e.g., using the multimycotoxin method, to ensure that you are only studying the effects of ZEN?
- Table 1: SD is very large in some cases, e.g. 33.13±48.53
Author Response
Congratulations on this excellent study and well written article based on an in vivo experiment with many results. The aim of this study was to determine whether low doses of zearalenone (ZEN) influence the carry-over of ZEN and its metabolites to intestinal tissues and the expression of CYP1A1 and GSTπ1 in the large intestine.
Thank you for your favorable review.
My comments:
- In such an experiment, why do not you analyze the feed for other mycotoxins, e.g., using the multimycotoxin method, to ensure that you are only studying the effects of ZEN?
- Little is known about the interactions of low doses with the macroorganism or other mycotoxins. As well as the knowledge at the level proposed by us is not very big. Therefore, we proposed a model of the experiment that is more unambiguous with the smallest possible number of factors introducing uncertainty.
- Table 1: SD is very large in some cases, e.g. 33.13±48.53
- This is not the only example where SD values are greater than mean concentration. This proves that the course of metabolic processes in pre-pubertal gilts is highly individual. Hence the averaging of the results for at least five animals in the experiment.
